# DNAzyme Sensor for the Detection of Ca^2+^ Using Resistive Pulse Sensing

**DOI:** 10.3390/s20205877

**Published:** 2020-10-17

**Authors:** Imogen Heaton, Mark Platt

**Affiliations:** Department of Chemistry, Loughborough University, Loughborough, Leicestershire LE11 3TU, UK; I.Heaton@lboro.ac.uk

**Keywords:** DNAzyme, aptamer, nanopore, resistive pulse sensor, metal ion sensor

## Abstract

DNAzymes are DNA oligonucleotides that can undergo a specific chemical reaction in the presence of a cofactor. Ribonucleases are a specific form of DNAzymes where a tertiary structure undergoes cleavage at a single ribonuclease site. The cleavage is highly specificity to co-factors, which makes them excellent sensor recognition elements. Monitoring the change in structure upon cleavage has given rise to many sensing strategies; here we present a simple and rapid method of following the reaction using resistive pulse sensors, RPS. To demonstrate this methodology, we present a sensor for Ca^2+^ ions in solution. A nanoparticle was functionalised with a Ca^2+^ DNAzyme, and it was possible to follow the cleavage and rearrangement of the DNA as the particles translocate the RPS. The binding of Ca^2+^ caused a conformation change in the DNAzyme, which was monitored as a change in translocation speed. A 30 min assay produced a linear response for Ca^2+^ between 1–9 μm, and extending the incubation time to 60 min allowed for a concentration as low as 0.3 μm. We demonstrate that the signal is specific to Ca^2+^ in the presence of other metal ions, and we can quantify Ca^2+^ in tap and pond water samples.

## 1. Introduction

Calcium (Ca^2+^) is one of the most abundant metals in the human body, making up to 2%wt of total human body weight, and plays a fundamental role in biological processes including secondary messengers critical for cell signalling, protein folding, and catalysis [1,2,3,4]. It has also been demonstrated that monitoring Ca^2+^ concentrations is important within the environment. High or very low concentrations within drinking water have been linked to problems with corrosion, scaling, and poor taste [5]. It also represents a hazard due to its high environmental mobility and bioavailability [6]. When unregulated, high concentrations of Ca^2+^ can cause various diseases such as hypercalcaemia [7] and heart disease, [8] while low levels can lead to deficiencies such as osteoporosis [9] and muscle and nerve tightening [10].

Many different methods have been developed for the detection of Ca^2+^ ions in solutions. Common methods include atomic absorption spectrometry [11,12], ion chromatography [13,14], and high-performance liquid chromatography [15]. While they are sensitive, e.g., the Limit of detection (LoD) for atomic absorption can be 0.005 μM [12], as well as selective and reliable, they are expensive, labour intensive, and unable to be taken on-site. Fluorometric methods have been widely acknowledged due to their advantageous short response times and high sensitivity; however, they too suffer complex operations, low detection throughput, and can also be problematic to apply on-site [16]. There remains a need for a technology that is relatively inexpensive, has short analysis times, is sensitive, and can be deployed in complex matrices.

An important component for any sensor is the recognition element, and thus, there have been components that have been developed to bind Ca^2+^ with high specificity that have been integrated into various sensing platforms, such as binding proteins [7,17,18], magnetic nanoparticles [19], and glycosphingolipds [20]. A new category of recognition ligands to heavy metals include DNA aptamers and DNAzymes. DNAzymes are DNA-based catalysts; all known DNAzymes have been selected through in vitro selection [8,21,22]. There are many DNAzymes that require divalent metal ions such as Pb^2+^ [23,24,25], Zn^2+^ [26,27], Cu^2+^ [28,29], and UO_2_^2+^ [30,31], for activity. DNAzymes have several advantages compared to enzymes commonly used within biosensing: they can easily be grafted onto surfaces, are more stable at ambient conditions, and can amplify detection due to their catalytic nature [32]. DNAzymes have been widely used as they have simple reaction conditions and significant changes in structure [33]. However, they usually require a tag or label to provide an analytical signal [3,34,35]. Resistive pulse sensing (RPS) is a sensing technology capable of probing changes in DNA structure [36,37]. RPS has a broad range of applications including material characterisation [38,39,40,41], quantification of DNA/peptide analyte interactions [42,43], and biosensing [44,45,46]. RPS measures speeds of nanocarriers as they translocate a nanopore, and through changes in nanocarrier speed, they are able to infer carrier binding with target analyte [47,48,49].

The integration of DNAzymes onto nanocarriers presents a new method of analysis, as the cleavage of the DNAzymes could be identified and characterised. RPS has demonstrated the ability to measure speed changes between double and single stranded DNA [36]. There have also been other studies that have utilised DNAzymes with a-hemolysin and solid-state nanopores [50,51,52]. Protein and solid-state nanopores offer the ability to distinguish between the DNAzyme and its cleavage products through changes in blockade signal. However, protein nanopores suffer from issues with stability and low reaction condition tolerance, while solid-state nanopores suffer from low signal-to-noise ratios and required large DNA strands to enable detection. To improve these limitations, carriers can be used to enable separation and preconcentration of samples, enhance the signal, and increase dynamic range and sensitivity of the assay [37,53].

Here, we present the use of a DNAzyme with RPS to detect Ca^2+^ ions in solution, using RPS’s ability to identify changes in structure. Previous studies have reported the cleavage of the DNAzyme by Ca^2+^ [3,54,55]; here, we were able to measure and take this further by inferring the rearrangement of the DNAzyme structure post cleavage. The catalytic nature of the DNAzyme allowed for the assay limits of detection to be tuned dependent on incubation times, as well as allowing quantification of Ca^2+^ when incubation times are kept constant. We were also able to demonstrate our assay is specific to Ca^2+^ observing no interferences from a mix of divalent ions. Finally, we showed the ability of our assay to be deployed in real environmental samples through the quantification of Ca^2+^ in both tap and pond water samples.

## 2. Experimental

### 2.1. Chemicals and Reagents

Calcium(II) chloride and potassium chloride were purchased from Fisher Scientific, UK. Carboxylated polystyrene particles were purchased from Bangs Laboratories, USA, and are denoted as CPC200 (mode diameter 210 nm, measured concentration 1 × 10^12^ particles mL^−1^). Nanopores were purchased from Izon Science Ltd., New Zealand. Reagents were prepared in deionised water (Elga PureLab), with a resistance of 15 MΩ cm. 2-(*N*-Morpholino)ethanesulfonic acid (MES) and 2-[4-(2-hydroxyethyl)piperazin-1-yl]ethanesulfonic acid (HEPES), Lithium Chloride, nickel sulphate hexahydrate, iron chloride, magnesium chloride, and TWEEN 20 were purchased from Sigma-Aldrich, UK. Streptavidin-modified magnetic particles, 120 nm, were purchased from Ademtech, France.

### 2.2. Custom DNA Oligonucleotides

Two custom DNA oligonucleotides were purchased from Sigma-Aldrich, in lyophilised form, and were purified using reverse-phase cartridge purification by the manufacturer. The two oligonucleotides ordered were: GCCATCTTTTCTCACAGCGTACTCGCTAAGGTTGTTAGTGACTCGTGAC (enzyme strand) and biotin-TCACGAGTCACTATRAGGAAGATGGCGAAA (substrate strand); they were diluted to 100 µM stock solutions using deionised water.

### 2.3. Particle Preparation

The DNAzyme complex was first formed by heating the substrate strand, 2.5 µL, with the enzyme strand, 5 µL, in 50 mM MES buffer (pH 6 with 25 mM LiCl) with 0.05% TWEEN 20 added, at 70 °C for 2 min followed by cooling. Once cool, streptavidin modified particles were added to the solution and left to bind for 30 min. Once bound, the particles were placed on a MagRack (Life Sciences) until a clear cluster of particles was seen; the buffer was then removed and replaced with 10 mM HEPES, pH 7.6, 50 mM LiCl, and a buffer with 0.05% TWEEN 20 was added.

The concentration of the particles in each assay was 2.5 × 10^9^ particles per mL. The suppliers data sheets give the binding capacity of 5665 pmol of biotin per mg of beads, which we used to calculate how much DNA is 100% coverage. In each assay, the number of particles remained the same and the DNA was always added in excess to ensure complete coverage of the carrier surface. We have shown how the number of the carriers and ligand density can affect the sensitivity in similar assays. This was not the subject of investigation here [37,42,48,53,56].

### 2.4. RPS Set Up

A qNano (Izon Science Ltd., New Zealand) was used to complete all the measurements for this study. A qNano uses data capturing software (Izon Control Suite v3.3) to record the particles as they traverse the pore. The lower fluid cell contained 80 µL of KCl solution and the upper fluid cell contained 40 µL sample solution. After each measurement was taken, the nanopore was cleaned by first rinsing the upper fluid cell with background buffer before the buffer was removed and replaced multiple times. Each time, a different pressure or vacuum was applied. The nanopore stretch was also varied wider and narrower; this was done until there were not any residual particles observed in the system, ensuring no cross contamination between the samples. The qNano was operated with a positive bias, i.e., the positive electrode in the lower fluid cell and the negative electrode in the upper fluid cell, so the particles traverse the pore towards the positive electrode unless otherwise stated. For all experiments, an NP200 nanopore was used to be able to analyse particles from 85–500 nm. More than 200 particles where measured for each sample, and a typical rate was 250–300 particles per min. To account for any manufacturing variation between pores, the baseline current was kept constant throughout all experiments, and the stretch was slightly changed to ensure this, with a maximum 10% difference in baseline between sample runs. Each pore was first characterised using the calibration particles so day-to-day differences could also be accounted for.

A typical workflow was 30 min incubation of the particles in the test solution before being vortexed and circa 2 min to acquire the data. If required, a magnetic separation step was included. After the incubation period, the samples were placed next to the magnet for 5 min or until a clear cluster of particles could be seen. The solution was then removed, and the particles resuspended before detection.

### 2.5. Saturation Point Testing

Particles where functionalised with different amounts of the DNA substrate strand added to determine the saturation point. Heating the substrate and enzymes strands together formed the DNAzyme before different amounts were added to the particles to determine DNAzyme forming and saturation point. Finally, post heating of the two DNA strands, they were left to cool in different places before being added to particles to determine the best cooling practices.

### 2.6. Binding Time

DNAzyme particles where functionalised as above, and Ca^2+^ was added to make 1 μM solutions. The samples were the placed on a rotary wheel, and at 5, 10, 20, and 30 min, an aliquot was taken out and vortexed before being analysed.

### 2.7. Metal Ion Interferences

Metal ion stocks were prepared from MgCl_2_, NiCl_2_, and FeCl_3_. These were added to DNAzyme functionalised particles to give 3 μM solutions. A mix was also prepared, which included Ca^2+^. Samples were left on a rotary wheel for 30 min before being vortexed and analysed.

### 2.8. Environmental Water Samples

Water samples were collected from a lap tap and an outdoor pond. DNAzyme particles were functionalised as above and added to the water samples. The samples were also spiked with Ca^2+^ to give a final concentration of 3 µM. They were then left on a rotary wheel for 30 min before being magnetically separated and resuspended in buffer before analysis.

### 2.9. Particle Speed

Particle speed through the pore was calculated from the pulse width. As particles translocate the nanopore, they produce a pulse. The maximum magnitude of this pulse is recorded as T_1.0_ as this magnitude decreases multiple time points are recorded correlating to T_0.9_, T_0.8_, T_0.7_, etc. Here, we used the reciprocal of the value at T_0.5_ (width of the pulse at 50% peak height) to determine relative particle speed. These values were then normalised to either calibration particles (CPC200s) or blank peptide functionalised particles were run on the same day. Using the same nanopore and experimental conditions, normalisation was done by running multiple calibration runs, getting an average setting of 1, and then dividing the sample data by this to get a ratio. This was done to account for any differences in measured speed between different pores and different days.

## 3. Results and Discussion

Within RPS experiments, each translocation of a carrier through the nanopore produces a pulse, such as in Figure 1a. The magnitude of the pulse, known as the pulse magnitude, ∆*i_p_*, is related to the volume of the carrier, and the width or full width half maximum, FWHM, of the pulse relates to its velocity [36]. In the absence of convection, the velocity of the carrier can be proportional to the surface charge or zeta potential of the carrier, assuming that electro osmosis remains constant [36,57]. Thus, RPS allows the quantity and structure of the DNA on the nanocarrier to be analysed [36].

The DNA sequences of the two DNA strands that make up the DNAzyme are shown in Table 1, and the two strands are labelled substrate and enzyme. The structure of the Ca^2+^ DNAzyme was previously reported by Yu et al. after systematic mutation to identify the optimal sequence. This is shown schematically in Figure 1bi [3]. The substrate strand has a single RNA linkage (rA) that operates as the cleavage site. Upon binding with two Ca^2+^ ions [55], the substrate strand cleaves and rearranges, changing from dsDNA to ssDNA, seen in Figure 1bii [3,4]. RPS can differentiate from dsDNA and ssDNA through changes in the nanocarrier translocation speed [36].

Previous work from our group has shown that the technique is cable of differentiating between the location and position of dsDNA on a mixed ss/dsDNA strand [36].

Here, the hypothesis was that the binding of Ca^2+^ ions to the DNA would cause a change in the tertiary structure that could be followed using RPS. The substrate strand binds to the nanocarrier via the biotin-avidin interaction. To confirm the immobilisation of the substrate strand on the carrier, varying concentrations were added to a consistent number of carriers, as in Figure 2a. At 120 nM, the velocity remains unchanged; note that more DNA may attach to the carrier’s surface, but the signal does not change above this concentration and we can infer that the particles are saturated with DNA. Due to the double stranded nature of the DNAzyme and the lower packing density of dsDNA on the carrier, it was expected that a lower concentration of DNAzyme would be required to cover the same surface area of carriers. A comparable experiment was carried out and the data are shown in Figure 2b. This shows a decrease in saturation point from 120 to 60 nM for dsDNA. It is important to note that the binding of the DNAzyme to the carriers did not result in a change in the pulse magnitude, indicating that changes in nanocarrier speed were down to differences in carrier charge, not aggregation or disaggregation of the particles, Appendix A. Alternative processes were also tested to ensure a high grafting density of the dsDNA on the carrier used different cooling cycles. When forming the DNAzyme complex, the two DNA strands are first added together, and the solution is heated to 70 °C before being cooled. To compare if the rate at which the solution cooled affected the packing, density samples were placed within the fridge (cold), room temperature (RT), heated (warm), and one was left in the dry bath (hot). The data presented in Appendix A compare the different speeds of the nanocarriers functionalised with the DNAzymes formed through the different cooling methods. As can be seen, although the differences are only slight, keeping the solution in a warm place to allow it to cool slowly led to the faster speed, indicating that more DNAzyme has been immobilised onto the carrier surface.

Studies have shown how the binding of Ca^2+^ to the DNAzyme complex cleaves a section of 15 base pairs of DNA, as in Figure 1bi,bii [3]. If the reported mechanism is correct, it was hypothesised that as the DNA cleaves, changing from dsDNA to ssDNA, it would result in the decrease in nanocarrier translocation speeds. [36] This theory was tested by monitoring the DNAzyme modified nanocarriers speeds at different time points after incubation with Ca^2+^. As can be seen in Figure 3ii, after the initial 5 min, we measured a significant decrease in nanocarrier speed, from 1 to 0.84 ms^−1^, before a rapid increase and plateauing after 10 min to 1.23 ms^−1^. We postulated that this increase in speed was due to the DNAzyme rearranging post cleavage by Ca^2+^, as the remaining attached DNA strand is extended, seen in Figure 1biii [58]. This increase in DNA length would increase particle speed significantly, and it correlates to previous studies [36]. The speed of the nanocarriers in the presence/absence of Ca^2+^ is shown in Appendix A. The velocity of the uncoated and ssDNA carriers remains unchanged with the addition of Ca^2+^, which illustrates that the addition of divalent ions does not change the electroosmotic and electrophoretic forces acting on the particles and Nanopore channel. We also interpret this change in velocity for the DNAzyme coated carriers as being specific to the cleavage mechanism.

Figure 4 demonstrates that the change in nanocarrier speed is proportional to the concentration of Ca^2+^ present in solution. Only the Ca^2+^ concentration was varied, as incubation times and DNA concentration remained the same. As shown in Figure 4a, as the concentration of Ca^2+^ increases, the speed of the nanocarriers increases. This constant translocation speed is stable over a large concentration range, from 3 μM, 1.23 ms^−1^, to 3000 μM, 1.27 ms^−1^, demonstrating the end point of the reaction, seen in Appendix A. The LOD for an assay run under these conditions, i.e., with 30 min incubation of DNAzyme particles in the Ca^2+^ solution, the LOD, calculated as 3 × STEY,X/gradient, is 1 μM. The saturation in velocity over 5 mm is due to the saturation of the binding sites. Whilst it may be possible for more Ca^2+^ to bind over this concentration, we are unable to resolve it. Due to the catalytic nature of DNAzymes, we also tested a lower concentration of Ca^2+^ ions, 0.3 μM, with varying incubation times from 30 to 150 min. As shown in Figure 4b, as incubation times increases, so did the nanocarrier speed until after 90 min when the speed becomes constant at 1.25 ms^−1^ and 1.23 ms^−1^ for 90 and 150 min, respectively. The change in translocation speed in the presence of Ca^2+^ was not due to particle aggregation or disaggregation, see Appendix A, which shows there was no measurable change in particle size. The change in translocation speed was reproducible over different batches of nanocarriers such as in Appendix A.

To demonstrate that the selectivity of our assay was dependent upon the Ca^2+^ binding to the DNAzyme, we incubated our DNAzyme functionalised particles with different metal ions. As the previous study had reported some binding with Mg^2+^ [55], Mg^2+^ was added at 3 µM and incubated for 30 min, and 5 mM was tested every hour over 6 h (Appendix A) to confirm that there was no interferences with our assay. As seen in Figure 5a, there were no inferences measured from any of the metals tested at 3 µM, and when present in a mix with Ca, the measured speed was within the expected range. Although some slight binding was overserved with 5 mM Mg^2+^, this was only measured after 4 h and not measured to a comparable speed as with the Ca^2+^ present at 3 µM, 1.09 vs. 1.21 ms^−1^ respectively. Additional divalent ions were also tested in the sensor, shown in Figure 5a. No change in translocation velocity was recorded. In a final experiment, all the HMI including Ca^2+^ was added to the DNAzyme modified particles, termed mix in Figure 5a. As can be seen, the translocation velocity is comparable to those recorded using Ca^2+^ alone.

Finally, to demonstrate the applicability of our assay in more complex sample matrices, we collected water samples from a tap in the laboratory and from a local pond on campus. To each sample, we added DNAzyme functionalised particles, and incubated for 30 min before magnetically separating them and resuspending in buffer to analyse. We also incubated the nanocarriers in water samples spiked with 3 μM Ca^2+^. As most calcium is present in water as calcium carbonate, we also incubated the nanocarriers in water samples spiked with 3 μM Ca^2+^ to show the ability of our assay to work in these different matrices without experiencing any interference from other analytes present. As shown in Figure 5b, the nanocarriers increased significantly in speed when compared to the blanks, indicating the ability of our assay to work in these different matrices without experiencing any interference from other analytes present.

## 4. Conclusions

Here we present RPS technologies inferring structure changes in the DNAzyme through differences in nanocarrier speed, enabling quantification of Ca^2+^ in solution. The cleavage of DNA post binding of the Ca^2+^ to the DNAzyme is measured through the increase in nanocarrier speed as it traverses the nanopore. The method works across a large concentration range, is tuneable through incubation times, and can work in environmental samples. The assay work flow from nanocarrier functionalisation to incubation, extraction, and quantification can be done in under an hour. 

## Figures and Tables

**Figure 1 sensors-20-05877-f001:**
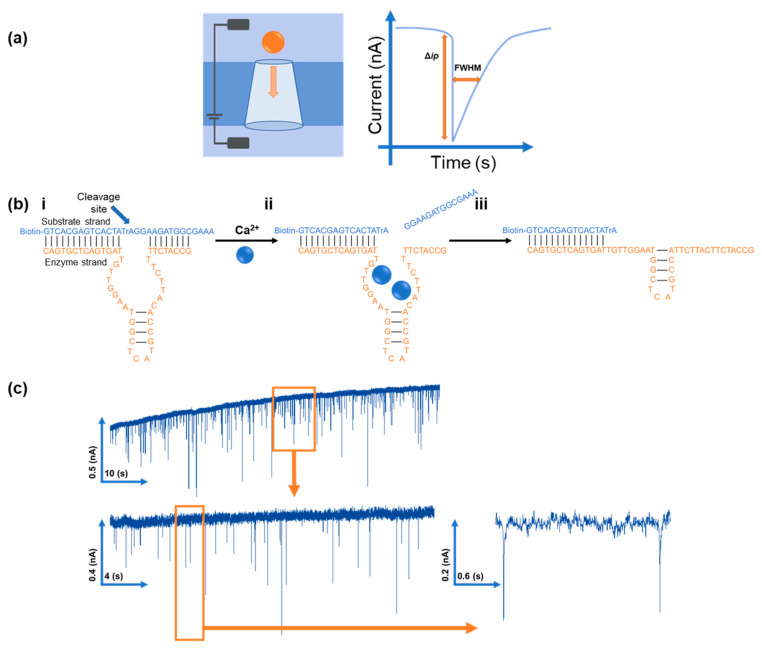
(**a**) Schematic of a particle traversing the RPS device and the signal produced. Blockade magnitude (Δ*i_p_*) and full width half-maximum (FWHM) are shown. (**bi**) Schematic of the DNAzyme once both strands have bound together, with the substrate strand (blue) and the enzyme strand (orange); (**bii**) binding with two Ca^2+^ ions and cleaving; (**biii**) rearrangement of DNAzyme post cleavage. (**c**) Example of a baseline current and blockade events caused by the carriers translocating the pore from 10 to 30 s, with the inset showing 17 to 21 s and two example pulses.

**Figure 2 sensors-20-05877-f002:**
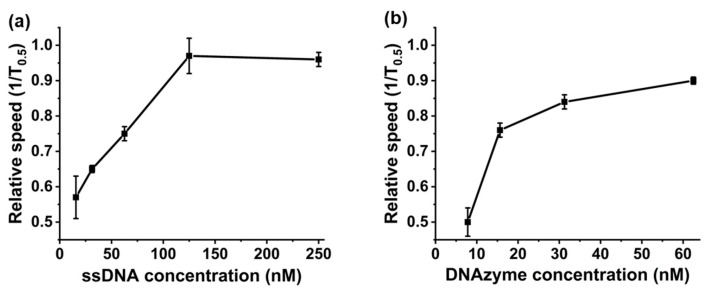
(**a**) Determining the concentration of substrate strand DNA required to saturate the nanocarriers through changes in carrier speed. (**b**) Determining if the DNAzyme complex had formed post heating through the differing saturation points of the substrate strand vs. the DNAzyme complex.

**Figure 3 sensors-20-05877-f003:**
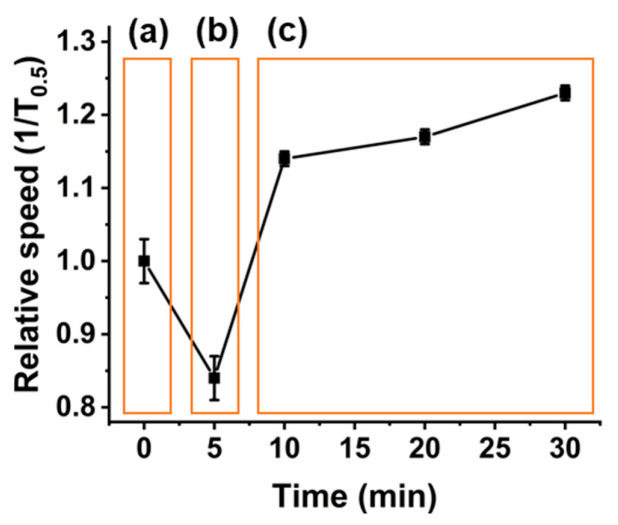
Measuring the change in the DNAzyme functionalised nanocarrier velocity versus different incubation times with Ca^2+^ ions; (**a**), blank DNAzyme complex on the bead, (**b**) cleavage of the DNAzyme from dsDNA to ssDNA, and (**c**) DNAzyme rearranging. Samples were run in triplicate.

**Figure 4 sensors-20-05877-f004:**
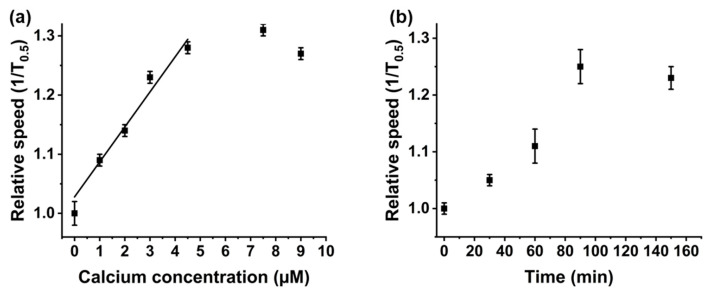
(**a**) Relative particle speeds normalised to blank DNAzyme functionalised particles, versus varying amounts of Ca^2+^ ions added from 1–9 μM, measured at 30 min. (**b**) 0.3 µM Ca^2+^ ions added, incubated from 30 to 150 min. Samples were run in triplicate, and between 250 and 300 particles were measured for each sample. Error bars are one standard deviation from the mean.

**Figure 5 sensors-20-05877-f005:**
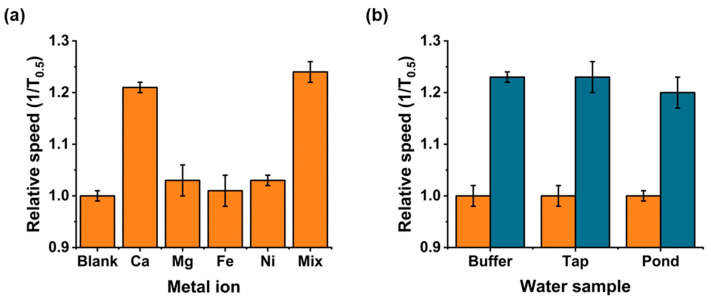
(**a**) Relative speed of the DNAzyme functionalised particles when incubated with Ca^2+^, Mg^2+^, Fe^3+^, and Ni^2+^ individually, and then when all present were mixed. All metals were present at 3 μM. (**b**) Environmental water samples, blank (orange), and spiked with 3 μM Ca^2+^ (green). Relative particle speeds normalised to blank DNAzyme functionalised particles, and they were run on the same day under the same conditions. Samples were run in triplicate, and between 250 and 300 particles were measured for each sample. Error bars are one standard deviation from the mean.

**Table 1 sensors-20-05877-t001:** Table of the two different DNAzyme strand sequences and lengths. The ribo-adenine is in the substrate strand denoted at rA, and it is here that the DNAzyme cleavages upon binding with calcium.

	DNA Sequence	
Substrate strand	[Btn]GTCACGAGTCACTATrAGGAAGATGGCGAAA	31 mer
Enzyme Strand	GCCATCTTTTCTCACAGCGTACTCGCTAAGGTTGTTAGTGACTCGTGAC	49 mer

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
