# Peer review of "DNAzyme Sensor for the Detection of Ca2+ Using Resistive Pulse Sensing"

_sensors, 2020, doi:10.3390/s20205877_

Round 1

Reviewer 1 Report

The manuscript by Heaton et al. deals with a Ca ion sensing system based on  DNAzyme using Resistive Pulse Sensing. Overall, the work looks interesting and the syetm shows high sensitivity towards Ca ion.

I would recommend the acceptance after the minor revision as stated below.

  1. The tile looks misleading as the system is applicable specifically to Ca not to the metal ions in general.
  2. The legend of Fig. 3 is confusing. There is only one fig. therefore 'a' is not necessary.
  3. Fig.4a: why the relative speed reaches a maximum at a certain Ca2+ concentration?
  4. Fig. 4 legend: (c) should be (b).
  5. Line 250: What 'constant speed' do authors refer to?
  6. Is there any variance in detection based on the anions associated with the Ca2+?

Reviewer 2 Report

In this manuscript, the authors reported an interesting sensing method for the detection of Ca2+ using a DNAzyme. The work was carefully performed and analyzed and I recommend publication of it after the following minor revision.

  1. The resolution of some figures are too low such as Figure 4. A linear fitting is needed in Figure 4A. What’s the detection limit for Ca2+? This needs to be calculated based on 3sigma and reported in the abstract as well (Not the lowest tested concentration).
  2. For the selectivity, only three competing metal ions were tested at one concentration. This needs to be expanded to more metals (at least Mg2+, Mn2+ and Cu2+ and Pb2+).
  3. Figure 3 has only one panel, no need to call (a).
  4. A control experiment showing DNAzyme cleavage actually happened would be important. This can be easily done by using an inactive mutant (e.g. changing a few important bases in the enzyme strand), or at least run a gel to show cleavage.

Reviewer 3 Report

Although the results are interesting, the lack of critical discussion of the results reduces the scientific soundness of the manuscript. However, there are some comments that need to be addressed before recommending the publication of the manuscript (major revision):

The introduction seems vague. Based on what is written now, it is hard to tell what the novelty is and what more precisely the practical applications are.

Materials and methods:

  • page 5, line 161: Cr3+ should be replaced by Ni2+ according to Figure 5;

Results:

  • Please include a suitable critical discussion of each of your results against scientific references on the subject. In parts, the results section feels like loose collections of statements without any/little comments or assessment.
  • Page 6, line 208: It seems that this data is reported in Figure 2a and not in Fig 2b. Moreover, "figure 2b" appears two times, the second being useless and with no coherency.
  • Page 7, lines 255-256: it should be clearly stated what the figures are reporting in the supplementary information.
  • Figure 5(a): there is no discussion about Fe, Ni or mix analysis in the text.
  • Figure 5(b): it is interesting to observe that there was no influence of the blank, either tap or pond water. How do you explain this as the ionic strength and pH considerably influence the 3D-conformation of DNA sequences?

Figures:

  • Figure 3: (a) should be deleted as there is no figure(b)

Conclusion: needs beefing up. It adds little value to the study.

  • Hence, it is stated that ”The method was works across a large concentration range” which is not the case – the difference between 3 μM (1.23 ms-1) and 3000 μM (1.27 ms-1) was of 0.04 ms-1 which is within SD.

References:

  • page 2, lines 42-43: it seems that the bibliography is not in accordance with the citations in the text; there's a shift of at least 2-3 references. Please verify them.
  • page 2, line 50: Ref. 18 presents an HPLC method, not a dipstick assay.
  • page 2, line 81: Self-citations should be as well checked. These references should be your previous studies as mentioned in the text.
  • a preprint of this manuscript was found in ChemRevix database https://doi.org/10.26434/chemrxiv.11993487.v1; the authors should mention it.

Other comments:

  • English syntax and grammar should be revised. Actually, there are many factual errors in the manuscript (i.e. page 1: lines 10,17; page 2: lines 37-39; page 2, line 62, page 3, line 112, page 9: 283-289, etc). The authors must realize an exhaustive proof-read before submitting the revised version.
  • Ca2+ - "2+" superscript in the abstract
  • Do not forget to describe the meaning of abbreviations the first time they are mentioned in the text.

Reviewer 4 Report

In the manuscript titled “DNAzyme Sensor for the Detection of Metal Ions using Resistive Pulse Sensing” by Heaton and Platt, authors described an application of the system described by Yu et al., (Reference 3 in manuscript) for the detection of calcium in milk using fluorescence methodologies, modified for the detection of calcium in water using differential resistive pulse sensing.

The manuscript appears interesting, the authors have expertise in the field of resistive pulse sensing studies, they recently published about the detection of heavy metal ions using peptide nanocarriers (which I suggest citing in the present manuscript).

However, some author statements are not sufficiently supported and some data are missing, such as the limit of detection, the linearity of measurements, or the sensitivity in real samples, necessary to demonstrate the possible application of this sensor. Thus, I suggest to considering manuscript after major revisions.

Moreover, I completely understand the difficulties about the COVID limitations that make hard to work in this time, but the manuscript presents several mistakes in text and figures, so please check for mistakes and for the use of inappropriate terms, such as (row 15) “cell signaling” instead of “cell signally” or (row 20) “can react” instead of “can reactor”, or row 146 circa 2 minutes. Also, check the pictures, such as graphical abstract and figure 1b, which are wrong about the substrate cleavage, the correct sequences are GTCACGAGTCACTATrA and GGAAGATGGCGAAA, not GTCACGAGTCACTATrA and AGGAAGATGGCGAAA, and also a wrong DNAzyme strand sequence is reported in iii.

Row 62 several references are reported, so probably the sentence is “Previous works have utilized ..”

Row 199-208. It is not clear the different ratio 2:1 between DNA substrate and DNAzyme for the production of dsDNA. Should be the ratio 1:1? Could be authors more explicit on the meaning of their results? Could the excess in ssDNA affect the sensor properties?

Figure S1 is not clear, could authors better describe their results.

About Figure S2 and row 211-220, I do not agree with the temperature classification of authors, because for scientific work, room temperature is taken to be in the range 20 to 25 °C (usually 25), differently from a human comfortable room temperature of 18 °C (even if 21 °C is the recommended living room temperature). Thus, I suggest to remove the data point at 18 °C, and not to stress too much the data obtained on temperature, because it does not seem to be sufficiently significant.

Row 228-230, it is not clear the reason that changing from dsDNA to ssDNA, we observed a decrease in nanocarrier translocation. The reference 38 is not appropriate to clarify this point.

Authors should better describe their hypothesis to explain the obtained results.

Row 235, Authors should better describe the change in DNA secondary structure which determine the strand extend. Anyway, considering that the size of molecule doesn't change, how affects the different spatial conformation  the speed of the nanocarriers? Could the process be explained by a decrease in speed because the binding of calcium to the DNAzyme could cause an increase in size and molecular volume? Could the subsequent speed increase because the lost in size and volume due to the cleavage and lost of a part of the substrate single strand? Could the released AGGAAGATGGCGAAA sequence bond again to the DNAzyme? If yes, how could this affect the measurement?

Authors should better describe this part.

Figure 4a demonstrate that, at 4-5 microM calcium concentrations, a plateau is already reached. This mean that higher concentrations of calcium cannot be determined by this system. Also, more points are necessary in the range 0-5 to demonstrate a good linearity for the calcium detection.

A limit of detection is required in order to know the lower calcium concentration detectable by the system.

Figure 4b indicates a strong addiction of the speed from the time. How could this dependence affect the sensitivity of detection. Should the assay require at least 90 minutes of measurements in order to have a good sensitivity? Authors should discuss this point.

Figure 5b only demonstrate that the system could work in other simple matrices, such as water. But not demonstrate the ability of the sensor to measure the calcium in real samples. Using as reference each of the water samples (buffer, tap and pond) for the measurement of relative speed, reset each effect of sample on the sensor in the measures when 3 microM calcium is added. More interesting is to set the buffer as the only reference-sample, and measure the relative speed in the other samples with and without addition of calcium.

Finally, I do not agree with the authors' statement that the method works across (a) a large concentration range, is (b) tunable through incubation times, and can ...

Because, (a) the range of linearity is not defined, calcium concentration higher than 6 microM shown the same relative speed value avoiding the determination of the ion concentrations; (b) strong dependence from the time suggest to use long measurement times to have a lower variability of the response, reducing the interest in the sensor.

Authors should improve their conclusions.

Round 2

Reviewer 2 Report

The authors have address most of my concerns. Just one minor point, the Ca2+ in the abstract need to be in superscript for the 2+.

Author Response

We thank the reviewer for pointing out this oversight, it has been updated throughout

Reviewer 3 Report

The revised version of the manuscript is much improved compared to the original version. Therefore, I recommend the publication of this manuscript Sensors Journal.

Author Response

Thank you.

Reviewer 4 Report

The authors improved the manuscript and addressed almost all of my concerns.

However, an important point remains about LOD. In fact, the authors indicate a LOD of 0.85 µM [please correct the m in M along the text (Abstract and Results)], while the measurements against time is carried out at 0.3 µM Ca2+ lower than the LOD value. Probably the LOD is lower, because the point at 0.3 µM Ca2+ appears to be still in a linear range in figure 4a. Could authors check this value?

I suggest to accept after solving this point.

Author Response

We have updated the lowercase m to M. The measurements carried out at 0.3 µM were done to demonstrate how, due to the catalytic nature of the DNAzyme, sensitivity of the assay can be improved through changing incubation times. The LOD value calculated here is based of the assay with 30 minutes of incubation time, although this can be improved to 0.3 µM by increasing the incubation time to 90 minutes.

To make this clearer we have updated the abstract to remove the LOD so it now reads:

A 30 min assay produced a linear response for Ca2+ between 1-9 um, and extending the incubation time to 60 mins allowed the concentration as low as 0.3 um.

Further we have changed the text to read:

‘This constant translocation speed is stable over a large concentration range, from 3 μM, 1.23 ms-1, to 3000 μM, 1.27 ms-1, demonstrating the end point of the reaction, figure S4. The LOD for an assay run under these conditions, i.e. with 30 minutes incubation of DNAzyme particles in the Ca2+ solution the LOD, calculated as 3*STEY,X/gradient, is 1 μM.’